# Scale Difference from the Impact of Disease Control on Pig Production Efficiency

**DOI:** 10.3390/ani12192647

**Published:** 2022-10-01

**Authors:** Yaguan Hu, Yanli Yu

**Affiliations:** School of Economics and Management, Ningxia University, Yinchuan 750021, China

**Keywords:** disease control, data envelopment analysis, productivity, threshold regression

## Abstract

**Simple Summary:**

This paper uses the data of fixed observation points in the country’s rural areas, the Data Envelopment Analysis model is used to calculate the production efficiency of pig farmers, which shows the efficiency level of pig breeding in the country. Then, the Tobit model is used to analyze that epidemic prevention and control has a positive effect on production efficiency. Finally, the threshold regression model was used to analyze the results: the effect of disease control on production efficiency was different under different breeding scales. When the breeding scale was less than 6.0002, the disease control had a negative impact on the production efficiency. When the breeding scale was greater than 12.9994, the disease control had a positive impact on the production efficiency. The research results of this paper not only help farmers to correctly understand the relationship between epidemic prevention and production efficiency, guide farmers to expand their breeding scale, and actively participate in epidemic prevention and control, but also provide a reference for the government to formulate policies according to different scales to guide farmers to carry out epidemic prevention and control.

**Abstract:**

Epidemic disease prevention plays a critical role in ensuring the healthy development of livestock farming, and the subjective willingness of breeders can be affected by the cost of epidemic disease prevention. To correct the misconception that farmers regard the cost of disease control as an ineffective cost, and to promote the healthy development of the pig breeding industry, our study employed the data envelopment analysis super-efficiency model and panel threshold regression model to evaluate the combination of the cost of epidemic disease prevention and swine productivity using data collected from 1998–2018 across 30 provinces in China. The following results were obtained. (1) The cost of epidemic disease prevention generated a non-linear on swine productivity when the swine farming scale was limited; (2) When the number of animals at the beginning of the year was less than 6.0002, swine productivity was impacted negatively; (3) When the number of animals at the beginning of the year ranged between 6.0002 and 12.9994, the impact was insignificant; (4) A strong correlation was observed between the expenses of epidemic disease prevention and animal productivity when the number of animals at the beginning of the year exceeded 12.9994. These results indicate that publicity should be enhanced to elucidate the combination of epidemic disease prevention and swine productivity among breeders. In addition, the government should introduce relevant policies to encourage the development of large-scale pig farming, such as subsidies for the construction of large-scale farms and insurance.

## 1. Introduction

Chinese inhabitants consume pork as a source of protein and their preferred meat product [1]. The livestock business in China is dominated by swine farming, a primary source of livelihood for the populace [2]. A country’s food security and social stability depend to a large extent on the healthy and sustained growth of the hog industry. China’s pork production increased from 39.660 million tons in 2000 to 52.96 million tons in 2021, a rise of 33.53 percent, according to data from the China Statistical Yearbook. Numerous animal diseases, such as blue ear disease, streptococcal disease, and Asian foot-and-mouth disease type I, impede the healthy growth of the swine sector and substantially erode the trust of swine farmers to continue producing swine. Disease outbreaks can potentially cost the nation as well as individual farmers and linked industries economically. An example is the August 2018 epidemic of African swine disease in China, which spread quickly to 31 provinces and resulted in the death and culling of 1.13 million live pigs, as well as 50–100 billion yuan in economic losses [3].

Despite the pertinent legislation and regulations implemented by the Chinese government to prevent and control epidemics, Chinese swine farmers are typically not sufficiently motivated to manage them [4]. Certain swine farmers are unable to scientifically and unbiasedly understand the relationship between disease management inputs and swine production efficiency because of inferior production conditions and low levels of medical expertise [5]. Animal breeding, farm management, hygienic cleaning, and waste disposal are important aspects of disease control in the swine industry, in addition to vaccination of farmed pigs [6]; therefore, farmers have to invest money, a lot of time, and energy. Certain farmers have long maintained that input costs and efficiency of swine production are inversely correlated [7]. However, certain farmers believe that increased disease control can enhance animal health, reduce farm morbidity, and boost farm productivity [8]. According to several studies, systematic and scientific disease management methods can increase the value of pigs, resulting in lower production costs, higher productivity, and new commercial potential for farms [9,10].

However, only one third of farmers practice standardized disease control [11]. Various factors, subjective and objective, prevent farmers from implementing scientific and standardized disease management strategies into practice [12,13]. For objective reasons, first and foremost, because disease control requires a lot of time and effort in addition to money, some farmers choose not to implement recommended disease control strategies [14,15]. Second, certain farmers lack the specialized knowledge necessary for disease control [16]. Third, the present rise in feed prices and low swine prices in the swine farming business have reduced farmers’ profitability, forcing them to cut down on their disease control expenses to accomplish cost control [17]. Fourth, the swine farming business has undergone certain “free-rider” phenomena because of the positive externalities of disease prevention [18]. Fifth, the willingness of farmers to prevent diseases is influenced by factors such as age, gender, and education [19,20]. Finally, a study indicated that disease control helps increase output efficiency when farming is extensive; therefore, small- and medium-scale farmers forego disease control investments to cut costs [21]. In terms of subjective reasons, first, farmers are excessively trusting of the government and veterinarians, and think that in the event of a disease epidemic, they can control it [22]. Second, farmers’ beliefs in disease control strategies and self-perception of competence determine whether they implement disease control [23], and farmers will implement disease control methods if they trust them and are confident in their ability to implement said methods. Third, farmers’ willingness to control illnesses is influenced by their capacity for risk perception [24].

Numerous insightful studies on swine breeding and disease prevention have been conducted from various perspectives, and these studies provide an excellent foundation for this study’s references. However, most studies have focused on the factors influencing farmers to carry out disease control from the objective level, but research on the cost of disease control and swine production efficiency from the subjective level is lacking. Scholars have disregarded the fact that the objective practice of disease control adopted by farmers is subjectively controlled. When farmers do not fully comprehend the relationship between disease control expenditures and productivity, they simply regard disease control expenses as sunk costs. However, when farmers correctly understand the relationship between expenditure on disease control and productivity, they proactively and actively participate in disease control. Although certain scholars have pointed out that the characteristics of farmers, management models, and feeding methods are different at different breeding scales, only few scholars have studied the relationship between epidemic prevention costs and pig production efficiency at different breeding scales.

Based on this, this study uses a large sample of data from fixed observation sites in rural areas across the country. The super-efficient slacks-based measure model was used to calculate swine production efficiency for each sample household. By constructing a Tobit model and a threshold regression model, we empirically analyze the effect of disease control costs on swine production efficiency under different scales to elucidate disease prevention and control among farmers. This can not only help farmers correctly understand the relationship between disease control and production efficiency, and prompt them to participate more actively in disease control, but also provide decision-making reference for the government to scientifically guide the development of pig industry on a large scale and adjust disease control policies according to different farming scales.

## 2. Methods and Data

### 2.1. Super-Efficient Slacks-Based Measure Model

Data envelopment analysis (DEA) was introduced by Charnes et al. [25]. It is an efficient method for evaluating the existence of multiple input and multiple output indicator decision units. DEA methods mainly include CCR, BCC, and SBM models. The CCR and BCC models are usually based on the premise of radial measurement, assuming that inputs and outputs vary in the same proportion, which is less common in practical conditions. Therefore, based on the traditional DEA model, Tone (2001) proposed a non-radial DEA model [26], the SBM model, in 2001, which eliminates the bias and influence caused by the differences in radial and angular choices and reflects the slack variables of input surplus and output deficit. The super-efficiency SBM model is an evolution of the SBM model, which further evaluates decision units with efficiency values exceeding 1 to obtain more accurate efficiency results as follows:(1)$$\left\{\begin{array}{l} \min\rho =\frac{1-\frac{1}{m}\sum_{i=1}^{m}s_{i}^{-}/x_{ik}}{1+\frac{1}{q}\sum_{i=1}^{q}s_{i=1}^{+}/y_{ik}}\\ x_{k}=X\lambda +s^{-}\\ y_{k}=Y\lambda +s^{+}\\ \lambda \geq 0,x^{+}\geq 0,x^{-}\geq 0 \end{array}\right\}$$
where ρ is the efficiency evaluation index, and $x_{k}$ and $y_{k}$ are the input and output vectors of the decision cell. $x_{ik}$ and $y_{ik}$ are the elements of the input and output vectors, respectively, *X* and *Y* are the input-output matrices, $S^{-}$ and $S^{+}$ denote the input-output slack variables, and λ is the column vector. When ρ  ≥ 1, the decision unit is efficient, when 0 ≤ ρ < 1, the decision unit needs to further improve the input-output ratio to achieve optimal efficiency.

### 2.2. Tobit Regression Model

Because the dependent variable swine breeding productivity is non-negative and homing is left at 0, regression analysis of the restricted dependent variable using the panel Tobit model can effectively circumvent estimation errors and improve regression accuracy. For the fixed-effects Tobit model, because a sufficient statistic of individual heterogeneity could not be found for conditional maximum likelihood estimation, as in the fixed-effects logit or count models, this study uses a random-effects panel Tobit for regression. To reduce the effects of heteroskedasticity and multi-collinearity on the regression analysis, all variables were regarded as logarithms in this study. The econometric model was constructed as follows:(2)$$y_{it}=\beta +\ln A_{it}+\ln B_{it}+\ln D_{it}+\ln E_{it}+\ln G_{it}+\ln H_{it}+\ln K_{it}+\varepsilon_{it}$$

In the above equation, *i* is the province, *t* is the year, β is the constant term, $\varepsilon_{it}$ is the error term, $y_{it}$ is the swine breeding productivity, $A_{it}$ is the village cadre household, $B_{it}$ is the party member household, $D_{it}$ is the age, $E_{it}$ is the number of years in school, $G_{it}$ is the agricultural training, $H_{it}$ is the epidemic input cost, and $K_{it}$ is the number of stocks at the beginning of the year.

### 2.3. Threshold Regression Model

Theoretically, the greater the scale of farming, the greater the uncertainty risk faced by farmers, and the greater the investment required for disease control. However, according to the law of diminishing returns to scale in economics, when the size of a swine farm is smaller than a specific value, the efficiency of swine production increases with investment in disease control costs, while other technical levels remain constant. When the size of a swine farm exceeds a specific value, the swine production efficiency decreases with investment in disease control. Therefore, the relationship between disease control costs and swine production efficiency is non-linear.

The basic form of the single threshold regression model given by Hansen [27] is
(3)$$Y_{it}=\xi_{i}+\mu_{1}X_{it}I(q_{it}\leq \gamma_{1})+\mu_{2}X_{it}I(\gamma_{1}≺q_{it}≺\gamma_{2})+\varepsilon_{it}$$

In Equation (3), $Y_{it}$ is the explanatory variable, $X_{it}$ is the independent variable, $q_{it}$ is the threshold variable, γ is the threshold estimate, and $\mu_{1}$ and $\mu_{2}$ are the parameters to be estimated. $\xi_{i}$ is the individual effect, $\varepsilon_{it}$ is the random perturbability, and $\varepsilon_{it}$: iid N(0,$\delta^{2}$). $\mu_{1}$ = $\mu_{2}$ indicates that there is no threshold effect between $Y_{it}$ and $X_{it}$ and vice versa. *I* (·) is an exponential function, and when $q_{it}$ and γ satisfy the equation condition, *I* (·) = 1; otherwise, it is zero.

As multiple thresholds may exist in real situations, a single threshold cannot objectively represent the relationship between variables. Therefore, Hansen generalized a single threshold to multiple thresholds. For instance, the dual threshold is modeled as shown in Equation (4):(4)$$Y_{it}=\xi_{i}+\mu_{1}X_{it}I(q_{it}\leq \gamma_{1})+\mu_{2}X_{it}I(\gamma_{1}≺q_{it}≺\gamma_{2})+\mu_{3}X_{it}I(q_{it}≻\gamma_{2})+\varepsilon_{it}$$

### 2.4. Indicator Selection

#### 2.4.1. Swine Farming Production Efficiency (Dependent Variable)

This study investigated the effect of disease control costs on hog production efficiency; therefore, hog production efficiency was selected as the explanatory variable. The super-efficient SBM model not only inherits many advantages of the traditional DEA model, but also effectively overcomes the defect where the maximum value of the efficiency measurement result of the traditional DEA model is 1. Therefore, this study directly selected the super-efficient SBM model to measure swine production efficiency in different years in each province of China. Software DEA-SOLVER Pro5.0.

A prerequisite for the use of a super-efficient SBM model is the definition of inputs and outputs. Based on existing studies, this study defines the number of farm household laborers, production and operation costs, and production house area as inputs. The operating income, number of farrowing rate per year, and production volume in the year are designated as outputs.

#### 2.4.2. Introduction of Independent Variables

This study examined the effects of disease control costs on swine production efficiency. Because the relationship between disease control cost and swine production efficiency is non-linear, it is important to consider the relationship between disease control cost and swine production efficiency under different farming scales. When the farm’s breeding scale is relatively large, the farm’s initial stocking number is also larger; when the farm’s breeding scale is smaller, the initial stocking number is also smaller. Therefore, this study used the number of stocks at the beginning of the year to measure the breeding scale. In the threshold regression, this study selects the cost of disease control as the core explanatory variable, the number of stocks at the beginning of the year as the threshold variable, and village cadre households, party members, age of household head, education level of household head, and agricultural training as the control variables in the threshold regression.

### 2.5. Data Sources

Data for 1998, 2003, 2008, 2013, and 2018 for each province and city used in this study were obtained from a large sample of microfarm household data from the National Rural Fixed Observation Point of the Rural Economic Research Center of the Ministry of Agriculture and Rural Affairs. They included provincial code, village code, year, village cadre household, party member household, number of family laborers, breeder’s age, breeder’s education level, production and operation cost, agricultural training, epidemic and disease control cost, area of the production house, operation income, number of stocks at the beginning of the year, number of slaughter houses throughout the year, production for the year, and total loss over the year.

A basic description of all variables in this study is shown in Table 1.

## 3. Result Analysis

### 3.1. Analysis of National Swine Production Efficiency Measurement Results

The total efficiency of swine production in all states, provinces, and cities is reflected in the integrated efficiency. In general, combined efficiency indicates the effectiveness of the province’s hog farming production efficiency if it exceeds 1, and if it is less than 1, it represents the province’s poor hog farming production efficiency. The trend in swine production efficiency in the eastern, central, and western provinces from 1998 to 2018 is shown in Figure 1, Figure 2 and Figure 3. As shown in Figure 1, the hog production efficiency of the eastern provinces decreased between 2003 and 2018 and increased gradually in all other years. This may be related to the SARS outbreak in 2003 and the African swine flu pandemic in 2018. Figure 2 and Figure 3 show that the epidemic had certain negative effects on swine production efficiency in the central and western regions in 2003 and 2018. Nevertheless, the decline was less severe than that in the eastern provinces. The eastern provinces are more affected by the diseases because they are geographically small, their farming areas are grouped together, their populations are mobile, and the disease spreads more swiftly there.

### 3.2. Spatial Distribution of Swine Production Efficiency in Provinces and Cities

The hog production efficiency values of 30 provinces and cities in China were classified into different levels using ArcGIS software. The Figure 4 below shows that there are more obvious regional differences in swine production efficiency in different regions in 30 provinces and cities across the country. These five graphs show that the nation’s overall swine farming production efficiency remained consistent and did not undergo significant swings in 1998, 2003, 2008, 2013, and 2018. However, in terms of spatial distribution, there is an obvious spatial agglomeration phenomenon in the hog-breeding industry. Although the production efficiency of hog breeding varies in different provinces and cities, from the perspective of spatial distribution, the production efficiency of hog breeding is generally high in the south and low in the north. The southern provinces, such as Guangdong, Hainan, Jiangxi, Anhui, Hunan, and Zhejiang, generally rank in the middle to upper levels, although swine breeding production efficiency varies. Similar to Shanxi, Henan, and Shandong, the production efficiency of grain-producing provinces generally rank in the middle to upper level. Although the production efficiency of northern provinces, such as Xinjiang, Inner Mongolia, Qinghai, Heilongjiang, and Jilin varies, it ranks within the moderate to low level. Studies show that relatively high literacy levels among farmers translate to increased proportions of wage income and relatively high pork consumption capacity. Furthermore, the lower the pork price index, the more optimized the transportation accessibility, and the higher the swine slaughtering capacity, intensity of environmental regulations, provincial swine farming scale, and production efficiency [28]. Southern provinces have a higher level of economic development, more optimized transportation access, more educated farmers, a denser population, higher consumption capacity for pork, and greater intensity of environmental regulations than northern provinces. Therefore, overall swine production efficiency in the southern provinces exceeded that in the northern provinces.

### 3.3. Tobit Regression of Swine Production Efficiency and Disease Control Cost

Before running the regression, the variables were tested for multicollinearity to rule out potential internal correlations. The results are shown in the Table 2 below. Each variable variance inflation factor is less than 3, indicating that the degree of collinearity between the variables is within a reasonable range.

The regression results obtained using Stata 16.0 software are shown in the following Table 3:

Every 1 percent increase in disease control cost led to a 0.013 percent increase in productivity in the swine industry. Investment in disease control costs will reduce disease outbreaks and spread, as well as the number of swine culled, resulting in increased productivity [29]. Furthermore, disease control can improve swine living conditions and farm management. Certain experiments have shown that if animals are healthy, comfortable, and well-nourished, their growth potential is maximized, resulting in maximum meat production [24], as well as improved pork quality and value [30]. Consequently, disease control costs can have a significant positive impact on swine production efficiency.

The number of years in the head of the swine-farming household has a significant negative effect on production efficiency. For every 1% increase in school years, the productivity of swine farming decreased by 0.005%. The number of years of schooling reflects the educational level of the farm household head. In China, owing to the traditional culture and unpopularity of science and technology, the swine farming industry is strenuous and undignified. Therefore, most relatively highly educated people will stay in the city and choose less strenuous and comfortable jobs. However, modern hog farming requires highly educated people to use and promote scientific farming methods and scientific disease control measures. Most of the staff in existing farms lack training and expertise [31]. Consequently, the number of years of schooling of farm heads negatively affects hog farming productivity.

Participation in agricultural training significantly increased hog-farming productivity. The productivity of swine farming increased by 0.037 percent when the number of participants in the agricultural training increased by 1%. The modern swine breeding industry necessitates extensive promotion of the use of new breeds, technologies, and techniques [32], as well as reasonable planning, scientific management, standardized disease control, and proper waste disposal [33]. Agricultural training can help swine farmers master this set of competencies, thereby allowing them to increase the efficiency of their swine production.

The number of swine in stock at the beginning of the year has a significant positive effect on the production efficiency of swine. For every 1% increase in the number of swine in stock at the beginning of the year, the production efficiency of the swine industry increased by 0.02%. The initial stock number of hogs reflects the farming scale. When the initial stock number of hogs was larger, this indicated that the slaughtering rate and slaughtering capacity of hogs in that year were higher. Robinson [34] found that the size of a swine farm had a significant positive effect on the technical efficiency of swine farming. Therefore, when the initial stock number of the year is large, the production efficiency of hog breeding is also high.

### 3.4. Robustness Tests

This study used the model replacement method to test the robustness of the derived Tobit regression results. A simple multiple linear regression model was selected for the regression analysis of the original data. The results show that the coefficients and significance of the core explanatory variables of epidemic control costs were consistent with the original results. The coefficients and significance of the other control variables remain largely consistent with the original results. This indicates that the Tobit 4 regression results are robust and credible (Table 4).

### 3.5. Threshold Regression Results

The results of the Tobit regression show that epidemic disease control cost has a significant effect on hog farming productivity. However, because the impact of epidemic disease control costs on hog farming productivity is multidimensional, its impact may be different from that of hog farmers, indicating a possible non-linear relationship between the variables. To determine whether there is a non-linear relationship between the variables, this study used a panel threshold regression model to test the above non-linear relationship.

#### 3.5.1. Threshold Effect Test

It has been found that disease control can effectively prevent the outbreak and spread of diseases and thus improve the efficiency of farming production [35]. At present, the domestic farming model is developing from small-scale free-range farming to large-scale farming [36]. Scaled farming has advantages in resource use efficiency and significantly improves farming efficiency [37]. Different management models, feeding methods, and development routes developed by different scale pig breeding sites are different. It has also been found that there is a spatial aggregation effect of large-scale pig farming, and different farming scales face different intensity of environmental regulations, risk coping ability and market opportunities [28]. Therefore, this paper puts forward the reasonable hypothesis that the effects produced by epidemic disease control are different in the face of different farming scales.

The threshold effect of the model was first tested by to Hansen (1999). The cost of disease control was used as the core explanatory variable and the number of stocks at the beginning of the year was used as the threshold variable. The model was estimated under the original hypothesis of the existence of single, double, and triple thresholds to obtain the F-statistic and *p*-value derived using the bootstrap method. The results show that the single threshold effect is significant at the 1% significance level, and the double threshold effect is significant at the 1% significance level. However, the triple threshold effect does not pass the significance test at the 10% level (see the table below). Therefore, the analysis in this study was based on the double-threshold model.

Furthermore, the two thresholds of the dual-threshold model are identified. Table 2 reports the estimates of the two thresholds and their corresponding 95% confidence intervals, and the likelihood ratio function plot in Figure 5 and Table 5 provides a clearer understanding of the process of threshold value estimation and confidence interval construction, where the dashed line is the critical value of the LR value at the 5% significance level and the area below the dashed line constitutes the 95% confidence interval of the threshold value. As shown in Figure 1, the LR statistic is close to zero within the 95% asymptotically valid confidence intervals (1.6409, 2.1242) and (2.4849, 2.6391), and the test results cannot reject the original hypothesis that the threshold estimates are consistent with their true values. Therefore, there is a double threshold effect in the model estimation, and the two threshold estimates are 1.7918 and 2.5649, respectively.

#### 3.5.2. Threshold Regression Result Analysis

Based on the obtained threshold estimates, this study focuses on the non-linear effect of the cost of disease control on the production efficiency of swine farming. The threshold effect test shows that the cost of disease control has a significant non-linear effect on the production efficiency of hog farming. Specifically, as the scale of swine farmers increases, the relationship between the cost of disease control and swine farming production efficiency is like a “U” shape. The specific performance of this feature is that when the number of swine farmers at the beginning of the year is less than the threshold value of 6.0002, the cost of epidemic disease prevention and control has a significant negative impact on production efficiency, with an impact coefficient of −0.003. When the number of stocks at the beginning of the year entered the threshold between 6.0002 and 12.9994, the influence coefficient rose to 0.004, but did not pass the 10% level of significance test. When the number of stocks at the beginning of the year exceeded the threshold value of 12.9994, the cost of epidemic disease prevention and control had a significantly positive impact on the production efficiency of swine breeding, with an impact coefficient of 0.029. Under different farming scales, the degree of influence of epidemic disease control costs on swine farming production efficiency is different, showing a significant double-threshold characteristic, which means that when the scale of swine farming is small, there is a significant negative correlation between epidemic disease control costs and swine production efficiency; when the scale of swine farming is in the middle, the correlation between epidemic disease control costs and swine farming production efficiency is not significant. When the scale of swine breeding is large, there is a significant positive correlation between the cost of disease control and production efficiency. Therefore, the effect of disease control costs on the production efficiency of swine breeding cannot be generalized and needs to be analyzed specifically according to the scale of breeding.

## 4. Discussion

Uninterrupted disease outbreaks have always been the main factor affecting the stable development of the pig industry, and reasonable and standardized disease prevention and control can effectively reduce disease outbreaks of diseases [38]. However, at present, certain farmers have an incorrect understanding of disease prevention and control. They regard disease prevention and control as ineffective inputs; therefore, their enthusiasm and initiative for disease prevention and control are weak [39]. To explore the relationship between epidemic prevention and control costs and pig production efficiency, this study uses DEA software to calculate the production efficiency of different farmers and builds a Tobit regression model to demonstrate the relationship between the two. According to existing research on farms at different scales, the characteristics of farmers, management methods, and feeding methods differ [40]. However, there is little research on the relationship between the cost of disease prevention and the production efficiency of pigs at different breeding scales. Therefore, this study explored the relationship between epidemic prevention and control costs and pig production efficiency at different breeding scales by constructing a threshold regression model and using the breeding scale as the threshold variable.

Because the efficiency value calculated by the DEA software ranges between 0 and 1, the Tobit regression model was selected for the analysis of the relationship between the cost of disease prevention and control, and the production efficiency of pigs. The regression results (Table 3) show that the cost of epidemic prevention and control had a significant positive impact on the production efficiency of pigs, with an impact coefficient of 0.013, which passed the 1% significance test. This is consistent with the conclusions of related studies [41]. Reasonable and standardized disease prevention and control can reduce the outbreak and spread of the disease, which not only reduces the number of pigs killed and culled owing to the infection but also reduces disease treatment costs [42]. Certain studies have found that the cost of disease prevention is significantly lower than the economic and death costs caused by disease outbreaks, and that the 10-year disease prevention cost is only approximately 2% of the disease treatment cost [43]. In the process of epidemic prevention and control, in addition to injecting vaccines into pigs, farmers will also pay attention to the hygiene and ventilation of the farm, so that the pigs can live in a comfortable and safe environment [44], to maximize the limited stimulation of the growth potential of pigs [45]. This can effectively improve the production and quality of pork [46]. Tonsor’s study found that consumers are willing to pay a 20% premium for high-quality pork [7]. Epidemic prevention and control can lead to reduced expenditure on epidemic disease treatment, increased pork production, improved pork quality, and increased income for farmers.

Through the threshold effect test (Table 6), this study found that when the breeding scale is used as the threshold variable, there is indeed a threshold effect between the cost of epidemic prevention and control and the production efficiency of pigs. This means that under different breeding scales, the impact of disease control costs on pig production efficiency is non-linear. The threshold regression results in Table 7 show that when the breeding scale is small (ln$K_{\mathrm{it}}\;$≤ 1.7918), the cost of epidemic prevention and control has a significant negative impact on the production efficiency of pigs, and the influence coefficient is −0.004, passing the significance at the 10% level test. This is consistent with the results obtained by previous studies [47,48]. Compared to large-scale pig farming, small-scale pig farming mostly exists in rural underdeveloped areas [49]. Owing to the scattered distribution of residents in remote mountainous areas, low population density, low population mobility, low intensity of government environmental regulations on pig breeding, and low incidence of diseases [50], pig farmers in remote mountainous areas have little awareness of epidemic prevention and control, and more of them regard epidemic prevention and control as a task assigned by the government, and regard epidemic prevention and control costs as ineffective inputs [51]. At the same time, to cope with the lack of supply in the live pig market, the government has introduced corresponding policies such as breeding subsidies, culling subsidies, and breeding insurance to encourage farmers to continue breeding [52]. Although the purchase of breeding insurance shows that farmers are aware of disease prevention and control [18], and the corresponding subsidy policy can also significantly increase the willingness of free-range farmers to disclose diseases [53], there are certain adverse incentives and moral hazard issues [54]. When the breeding scale is small, the breeding subsidies, culling subsidies, and insurance compensation obtained after the occurrence of epidemic diseases are sufficient to offset the cost of epidemic disease treatment, which makes some farmers rely excessively on various subsidies and choose to reduce the input of epidemic disease prevention and control costs [55].

When the breeding scale was large (ln$K_{\mathrm{it}}$ > 2.5649), the cost of epidemic disease prevention and control had a significant positive impact on the production efficiency of pigs, with an impact coefficient of 0.028, which passed the 1% significance test. This is consistent with the conclusions of related literature [56]. As large-scale farming has the advantages of high production efficiency, strong disease prevention and control ability, convenient management, and suitability for the use of large-scale production equipment [57], it can achieve optimal utilization of capital and factors. Presently, owing to rising labor costs, the increasing intensity of environmental regulations, and the pressure of disease prevention and control, backyard farmers are accelerating their withdrawal, the structure of pig farming is changing from pyramid and spindle to dumbbell, and large-scale pig farming is developing rapidly [58]. When the scale of breeding is large, the cost input of farmers, such as feed, workshops, labor costs, transportation costs, manure treatment costs, management costs, and epidemic prevention and control costs, is also large. Simultaneously, there are additional requirements for environmental policies [59,60]. When an epidemic occurs, although culling subsidies and breeding insurance will compensate farmers to a certain extent, they cannot compensate for all losses and farmers still suffer heavy losses. In addition, large-scale farming pays more attention to animal welfare. On the one hand, it ensures that pigs live in a comfortable and safe environment; on the other hand, it can stimulate optimal growth potential of pigs and yield higher quality pork [61]. High-quality pork will be more expensive and therefore more profitable for farmers, and present additional opportunities for farmers, such as high sales rates [62]. Therefore, when the breeding scale is large, disease control can increase the efficiency of pig production by reducing the expenditure on disease treatment and increasing income.

The conclusion of this study not only makes farmers correctly understand the relationship between disease control and farming production efficiency, but also can motivate farmers to develop large-scale production. It can also help the government to formulate epidemic disease control policies and farming assistance policies according to different farming scales to promote the healthy and stable development of the pig industry.

## 5. Conclusions and Policy Recommendation

This study examined the relationship between disease control costs and swine production efficiency from the perspective of increasing the efficiency of swine production. Using relevant data obtained from 30 provinces in China from 1998 to 2018, this relationship was empirically tested using the super-efficiency DEA model and panel threshold regression model. The results showed that the cost of epidemic prevention and control had a significant impact on the production efficiency of pigs. Further, this paper used the threshold regression model to find that this effect varies with the size of the pig farm. When the breeding scale was small (ln$K_{\mathrm{it}}$ ≤ 1.7918), the cost of epidemic prevention and control had a significant negative impact on the production efficiency of pigs. When the breeding scale was large (ln$K_{\mathrm{it}}$ > 2.5649), the cost of epidemic prevention and control had a significant positive impact on pig production efficiency.

Furthermore, the following policy recommendations are drawn from the study findings. First, the government should strengthen the publicity and promotion of the necessity of epidemic disease prevention and control policies so that farmers can fundamentally understand the relationship between epidemic disease prevention and pig production efficiency and guide farmers to carry out orderly, scientific, and reasonable disease control actively and consciously. Second, an epidemic prevention and control model should be developed in line with the country’s national conditions. Different epidemic prevention and control measures should be adopted according to the region, culture, climate, and breeding scale. Finally, the advantages of high yield, low cost, and convenient management of large-scale breeding should be identified and relevant policies, such as feed subsidies and insurance subsidies, introduced to encourage the large-scale development of pig breeding. It should be noted that this paper has learned that there are scale differences in the effects of disease control on pig production efficiency, but the specific reasons for the scale differences between the two have not been explored in this paper. Therefore, in further studies, we can conduct in-depth research on the specific reasons, analyze and explore the multi-level factors that lead to different effects of disease control at different scales, and promote the healthy and stable development of hog industry.

## Figures and Tables

**Figure 1 animals-12-02647-f001:**
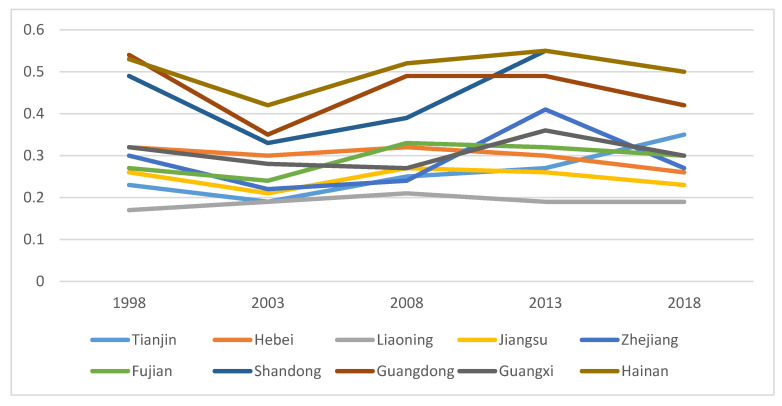
Productivity in Eastern Provinces during 1998–2018.

**Figure 2 animals-12-02647-f002:**
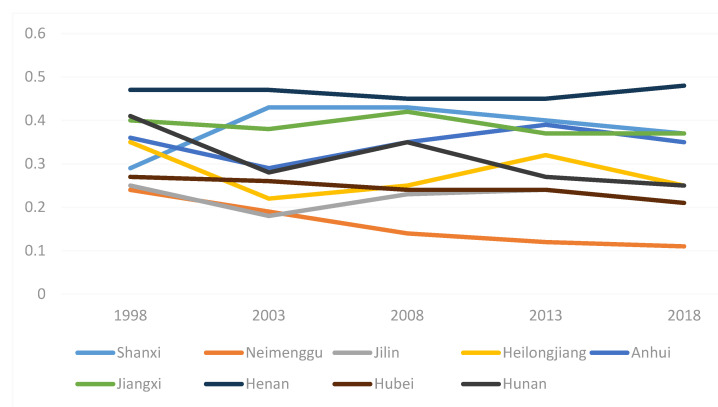
Productivity in Central Provinces 1998–2018.

**Figure 3 animals-12-02647-f003:**
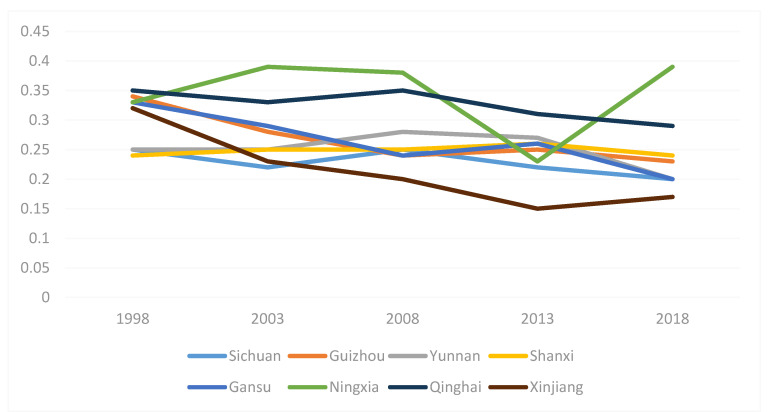
Productivity in Western Provinces during 1998–2018.

**Figure 4 animals-12-02647-f004:**
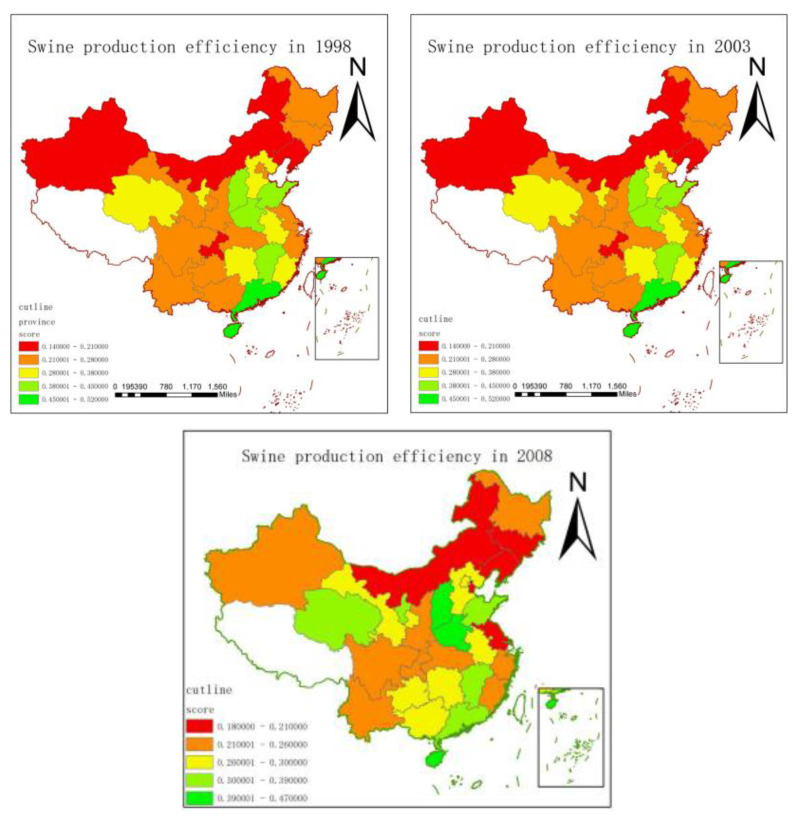
Swine production efficiency for 1998–2018.

**Figure 5 animals-12-02647-f005:**
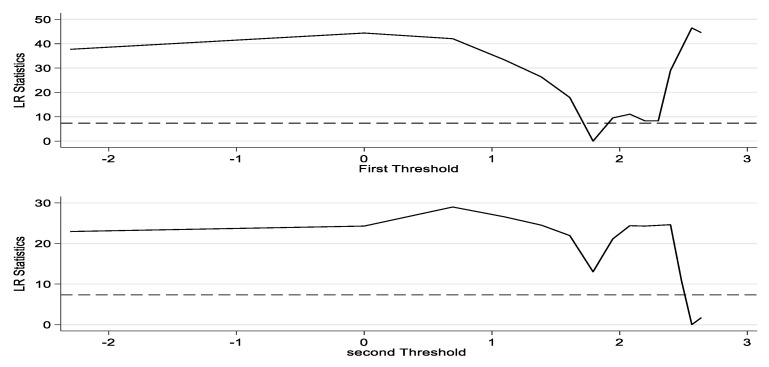
Threshold estimates and their confidence intervals.

**Table 1 animals-12-02647-t001:** Variables and results of descriptive statistical analysis.

Measurement	Variable	Meaning	Mean	Std. Dev	Min	Max
Swine production efficiency	$y_{it}$	Epidemic cost input of the farmer in province i in year t	0.309	0.193	0.03	1
Rural cadre households	$A_{it}$	Whether the farmer in province i is a village cadre in year t	1.955	0.207	1	2
Party member households	$B_{it}$	Whether the farmer in province i is a party member in year t	1.851	0.356	1	2
Number of laborers	$C_{it}$	Number of laborers in year t of farming households in province i	2.783	1.039	0	6
Age	*D_it_*	Age of head of household in year t of farming household in province i	53.712	10.387	29	80
Years in school	*E_it_*	Years in school for head of household in year t of farming households in province i	6.386	2.676	0	12
Production and operation costs	$F_{\mathrm{it}}$	Farming expenditure of farmers in province i in year t	4569.717	5622.458	25	20,985
Agricultural Training	$G_{it}$	Whether farmers in province i participated in agricultural training in year t	1.567	0.496	1	2
Disease control costs	$H_{it}$	Cost of disease control spent by farmers in province i in year t	132.560	125.665	3	400
Production room area	$I_{\mathrm{it}}$	Area of the production house of the farmer in province i in year t	33.595	34.868	0	200
Operating income	$J_{it}$	Farming income of farmers in province i in year t	7194.259	8443.411	20	31,379
Number of stock at the beginning of the year	$K_{it}$	Number of stock at the beginning of year t for farmers in province i	4.287	4.517	1	15
Number of slaughter throughout the year	$L_{\mathrm{it}}$	Number of farms slaughtered in year t by farmers in province i	7.349	10.011	1	33
Production for the year	$M_{it}$	Intra-year production of farmers in province i in year t	421.465	527.124	1	2100

**Table 2 animals-12-02647-t002:** Covariance test results.

Variable	Code	VIF	1/VIF
Ln (Rural cadre households)	$A_{\mathrm{it}}$	1.17	0.8528
Ln (Party member households)	$B_{\mathrm{it}}$	1.19	0.838
Ln (Number of laborers)	$C_{\mathrm{it}}$	1.02	0.979
Ln (Age)	$D_{\mathrm{it}}$	1.17	0.8565
Ln (Years in school)	$E_{\mathrm{it}}$	1.12	0.8897
Ln (Production and operation costs)	$F_{\mathrm{it}}$	2.55	0.3921
Ln (Agricultural Training)	$G_{\mathrm{it}}$	1.14	0.8736
Ln (Disease control costs)	$H_{\mathrm{it}}$	1.56	0.6415
Ln (Production room area)	$I_{\mathrm{it}}$	1.10	0.9093
Ln (Operating income)	$J_{\mathrm{it}}$	2.22	0.4506
Ln (Number of stock at the beginning of the year)	$K_{\mathrm{it}}$	1.54	0.6492
Ln (Number of slaughter during the year)	$L_{\mathrm{it}}$	2.05	0.4877
Ln (Production during the year)	$M_{\mathrm{it}}$	1.25	0.7995
Mean VIF		1.47	

**Table 3 animals-12-02647-t003:** Descriptive statistics of the variables.

Explanatory Variables	Coefficient	Standard Deviation	*p* > |t|
ln$H_{\mathrm{it}}$	$0.013^{\;***}$	0.002	0.000
${\mathrm{lnA}}_{\mathrm{it}}$	0.024	0.015	0.112
ln$B_{\mathrm{it}}$	0.01	0.009	0.257
ln$D_{\mathrm{it}}$	0.008	0.011	0.74
ln$E_{\mathrm{it}}$	$-0.005{\;}^{**}$	0.002	0.025
ln$G_{\mathrm{it}}$	$0.037{\;}^{***}$	0.006	0.000
ln$K_{\mathrm{it}}$	$0.02{\;}^{***}$	0.002	0.000
Constant term	$0.182{\;}^{***}$	0.046	0.000

Note: $A_{\mathrm{it}},{\;B}_{\mathrm{it}},{\;D}_{\mathrm{it}},{\;E}_{\mathrm{it}},{\;G}_{\mathrm{it}},{\;H}_{\mathrm{it}},{\;K}_{\mathrm{it}}$, respectively, are village cadre households, party member households, age, number of years in school, agricultural training, cost of disease control, and number of stocks at the beginning of the year, respectively; |t| is the confidence interval; *p* is the confidence level; **,*** indicates passing the significance test at 5%, and 1% levels, respectively.

**Table 4 animals-12-02647-t004:** Stability test.

Explanatory Variables	Coefficient	Standard Deviation	*p* > |t|
ln$H_{\mathrm{it}}$	$0.013{\;}^{***}$	0.002	0.000
${\mathrm{lnA}}_{\mathrm{it}}$	0.024	0.015	0.101
ln$B_{\mathrm{it}}$	0.01	0.009	0.255
ln$D_{\mathrm{it}}$	0.008	0.011	0.463
ln$E_{\mathrm{it}}$	$-0.005{\;}^{**}$	0.002	0.023
ln$G_{\mathrm{it}}$	$\;0.036{\;}^{***}$	0.006	0.000
ln$K_{\mathrm{it}}$	$0.02{\;}^{***}$	0.002	0.000
Constant term	$0.183^{\;***}$	0.045	0.000

Note: **,*** indicates passing the significance test at 5%, and 1% levels, respectively.

**Table 5 animals-12-02647-t005:** Threshold estimates and their confidence intervals.

	Threshold Estimates	H_it_ Corresponding Value	95% Confidence Interval
First threshold	1.7918	6.0002	[1.6094, 2.1242]
Second threshold	2.5649	12.9994	[2.4849, 2.6391]

**Table 6 animals-12-02647-t006:** Results of the threshold effect test.

Models	F-Statistic Value	*p*-Value	1% Critical Value	5% Critical Value	10% Critical Value
Single Threshold	$540.39^{\;***}$	0.0000	12.3728	9.5033	8.1401
Double Threshold	$35.12{\;}^{***}$	0.0000	12.3354	8.1655	7.6050
Triple Threshold	10.20	0.2800	28.9841	21.9620	16.1677

Note:*** indicate significance at the 1% levels, respectively; *p*-values and critical values are the results obtained from repeated sampling 400 times using the bootstrap method.

**Table 7 animals-12-02647-t007:** Results of estimating parameters of the panel threshold model.

Variables	Regression Coefficient	t-Value
${\mathrm{lnA}}_{\mathrm{it}}$	0.014	0.84
ln$B_{\mathrm{it}}$	0.011	1.13
ln$D_{\mathrm{it}}$	0.013	1.12
ln$E_{\mathrm{it}}$	$-0.006^{\;**}$	−2.19
ln$G_{\mathrm{it}}$	$0.038^{\;***}$	5.85
ln$H_{\mathrm{it}}\cdot 1$(ln$K_{\mathrm{it}}\;$≤ 1.7918)	−0.004 *	−1.71
ln$H_{\mathrm{it}}\cdot 1$(1.7981 < ln$K_{\mathrm{it}\;}$≤ 2.5649)	0.004	1.40
ln$H_{\mathrm{it}}\cdot 1$(ln$K_{\mathrm{it}\;}$> 2.5649)	$0.028^{\;***}$	12.34

Note: ***, **, * represent passing significance tests at the 1%, 5%, and 10% levels, respectively, and significance levels are marked with robust standard deviation estimates.

## Data Availability

The data presented in this study are available on request from the first author.

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
