# Peer review of "Scale Difference from the Impact of Disease Control on Pig Production Efficiency"

_animals, 2022, doi:10.3390/ani12192647_

Round 1
Reviewer 1 Report
Comments to the authors
Abstract and Introduction: you should report clearly the aim of the study
L13: .. cost, and to promote..
L24: .. should be enhanced to elucidate..
L88:… only 89 few scholars…
L100: provide Ethical Notes for your study, including the approval number by the Ethics Committee
L166: farrowing rate per year…
L186: .. the production house..
L207: diseases…
L228: .. generally rank in the middle to upper level..
L231: .. of wage income and relatively high..
L233: .. accessibility, and the higher..
L236: .. a denser population, higher consumption….
L389: Tonsor's study..
L447: .. had a significant
L457-458: … guide farmers to carry out orderly, scientific, and reasonable disease control actively and consciously
Reviewer 2 Report
1. Eliminate multiple references. After that please check the manuscript thoroughly and eliminate all the lumps in the manuscript. This should be done by characterising each reference individually. This can be done by mentioning 1 or 2 phrases per reference to show how it is different from the others and why it deserves mentioning.
2. The conclusion part should be more refined to make the findings and contributions of the paper clearer. Furthermore, please note the difference between the conclusions and abstract.
3. The method and data sources were clearly and exhaustively outlined. The methods have been clearly described, ensuring that qualified researchers can replicate your results. However, you should consider mentioning any assumptions made in the methods and experiment sections and how they may affect outcomes.
4. The results were explained well and key findings were adequately represented through figures and tables. To highlight the contributions of your study, consider making comparisons between your key findings and those of related studies, and mentioning the research gaps filled by the study findings.
5. The conclusion summarizes the main findings of this study in a cogent manner. To make this section more robust, please reiterate the objective of the study and how it was met. In addition, please mention the scope for further study and outline the limitations of the study in this section.
6. The use of too many words to convey one idea can muddle the message and divert the reader’s attention. Therefore, in writing, especially academic writing, ideas need to be conveyed as concisely as possible. One way of doing this is to use concise alternatives to phrases. For example, the phrase “all over the world” can be replaced with the word “globally” or “worldwide.”
